# Managing hypertension in frail oldest-old— The role of guideline use by general practitioners from 29 countries

Céline Roulet[1], Zsofia Rozsnyai[1], Katharina Tabea Jungo[1], Milly A. van der Ploeg[2], Carmen Floriani[1], Donata Kurpas[3], Shlomo Vinker[4], Sanda Kreitmayer Pestic[5], Ferdinando Petrazzuoli[6], Kathryn Hoffmann[7], Rita P. A. Viegas[8], Christian Mallen[9], Athina Tatsioni[10], Hubert Maisonneuve[11], Claire Collins[12], Heidrun Lingner[13], Rosy Tsopra[14,15], Yolanda Mueller[16], Rosalinde K. E. Poortvliet[2], Jacobijn Gussekloo[2,17], Sven Streit[1] *

**1** Institute of Primary Health Care (BIHAM), University of Bern, Bern, Switzerland, **2** Department of Public Health and Primary Care, Leiden University Medical Center, Leiden, The Netherlands, **3** Family Medicine Department, Wroclaw Medical University, Wroclaw, Poland, **4** Sackler Faculty of Medicine, Tel Aviv University, Tel Aviv, Israel, **5** Family Medicine Department, University of Tuzla, Tuzla, Bosnia and Herzegovina, **6** Department of Clinical Sciences, Center for Primary Health Care Research, Lund University, Malmö, Sweden, **7** Department of General Practice and Family Medicine, Medical University of Vienna, Vienna, Austria, **8** Department of Family Medicine, NOVA Medical School, Lisbon, Portugal, **9** Primary Care and Health Sciences, Keele University, Newcastle, United Kingdom, **10** Research Unit for General Medicine and Primary Health Care, University of Ioannina, Ioannina, Greece, **11** Primary Care Unit, University of Geneva, Geneva, Switzerland, **12** Irish College of General Practitioners, Dublin, Ireland, **13** Center for Public Health and Healthcare, Hannover Medical School, Hannover, Germany, **14** INSERM, Research center in Information Science to Support Personalized Medicine, University Paris Descartes and University Sorbonne Paris Cité, Paris, France, **15** INSERM, LTSI Team Health Big Data, University of Rennes, Rennes, France, **16** Department of Family Medicine, Center for primary Care and Public Health (Unisanté), University of Lausanne, Lausanne, Switzerland, **17** Department of Gerontology and Geriatrics, Leiden University, Leiden, The Netherlands

☯ These authors contributed equally to this work.

* sven.streit@biham.unibe.ch

**Data Availability Statement:** All relevant data are within the manuscript and its Supporting Information files.

## Abstract

### Background

The best management of hypertension in frail oldest-old ($\geq$80 years of age) remains unclear and we still lack guidelines that provide specific recommendations. Our study aims to investigate guideline use in general practitioners (GPs) and to examine if guideline use relates to different decisions when managing hypertension in frail oldest-old.

### Design/Setting

Cross-sectional study among currently active GPs from 29 countries using a case-vignettes survey.

### Methods

GPs participated in a survey with case-vignettes of frail oldest-olds varying in systolic blood pressure (SBP) levels and cardiovascular disease (CVD). GPs from 26 European countries and from Brazil, Israel and New Zealand were invited. We compared the percentage of GPs

**Funding:** Prof. Streit's research was supported by grants (P2BEP3_165353) from the Swiss National Science Foundation (SNF) and the Gottfried and Julia Bangerter-Rhyner Foundation, Switzerland. Christian Mallen is funded by the NIHR Collaborations for Leadership in Applied Health Research and Care West Midlands, the NIHR SPCR and a NIHR Research Professorship in General Practice, (NIHR-RP-2014-04-026). The views expressed are those of the authors and not necessarily those of the National Health Service, the NIHR or the Department of Health and Social Care. The funders had no role in study design, data collection and analysis, decision to publish, or preparation of the manuscript.

**Competing interests:** The authors declare that they have no competing interests.

reporting using guidelines per country and further stratified on the most frequently mentioned guidelines. To adjust for patient characteristics (SBP, CVD and GPs' sex, years of experience, prevalence of oldest-old and location of their practice), we used a mixed-effects regression model accounting for clustering within countries.

## Results

Overall, 2,543 GPs from 29 countries were included. 59.4% of them reported to use guidelines. Higher guideline use was found in female (p = 0.031) and less-experienced GPs (p<0.001). Across countries, we found a large variation in self-reported guideline use, ranging from 25% to 90% of the GPs, but there was no difference in hypertension treatment decisions in frail oldest-old patients between GPs that did not use and GPs that used guidelines, irrespective of the guidelines they used.

## Conclusion

Many GPs reported using guidelines to manage hypertension in frail oldest-old patients, however guideline users did not decide differently from non-users concerning hypertension treatment decisions. Instead of focusing on the fact if GPs use guidelines or not, we as a scientific community should put an emphasis on what guidelines suggest in frail and oldest-old patients.

## Introduction

Hypertension is highly prevalent worldwide especially in oldest-old. In primary care, general practitioners (GPs) are paramount to decide on optimal blood pressure goals. However how best to treat hypertension in oldest-old (80 years or older) patients, especially those who are frail, is still an open question [1]. This population group is rarely the subject of specific recommendations in currently available guidelines for treating hypertension.

Oldest-old patients are a rapidly increasing segment of the population, and GPs see more and more of them [2]. These patients are a heterogeneous group. Some are healthy, while others are frail and live with multiple complex medical conditions. Despite the increase, this population is widely excluded from clinical trials, particularly from hypertension trials [3]. Most studies apply in fact very strict criteria excluding patients with other diseases than the condition under study, which reduces the generalizability of the results [4]. This statement is especially relevant in a primary care setting where over two thirds of patients over 50 of age have more than one chronic disease [5].

Treating hypertension effectively decreases cardiovascular risk factors in the general population [6], but there are no reliable data whether it is also the best treatment-strategy in the oldest-old. Whilst some trials suggested that lowering blood pressure benefits this group [7], most of these trials included only fit members of that age group. Meanwhile observational studies reported that low systolic blood pressure was associated with an increase in all-cause mortality in the oldest-old [8–10].

Many different hypertension guidelines are available. GPs therefore have to choose between regional, national, continental or international recommendations. As illustration, a well-known European guideline is edited by the European Society of Cardiology (ESC) with already four updated versions, the current one published in 2018 [11]. The National Institute for

Health and Care Excellence (NICE) also provides some national recommendations, which are popular among English GPs [12]. Another national guideline, developed by the Dutch College of GPs (NHG), is implemented in the Netherlands [13].

In this study we aim to assess if GPs used guidelines when deciding on how to treat hypertension in their oldest-old and frail patients, and if guideline-users decide differently from non-users.

## Methods

### Design

We conducted a re-analysis of data from the cross-sectional case-vignettes study called 'Antihypertensive TreaTmENT In Very Elderly' (ATTENTIVE) [14].

### Setting

We organized a network of 'national coordinators' (mostly one per country) through national and European GP's organizations. The role of the national coordinator was to seek ethical approval for the local data collection (if applicable), supervise translation of the survey and send out the survey and reminders to their GP network(s). The surveys were distributed from spring to summer 2016. The detailed design of the ATTENTIVE Study has been described previously [14,15].

### Ethical considerations

Our study accords with the ethical principles of the Declaration of Helsinki [16]. The GPs' responses to the survey served as their informed consent. Because the survey was anonymous, there was no need to seek ethical approval in most countries. The ethics committees of Brazil and Switzerland specifically waived the requirement. We sought and obtained approval from the ethics committee of Auckland University in New Zealand.

### Participants

The only inclusion criteria to participate in the ATTENTIVE study, was to be currently active as a GP. We excluded GPs who were not practicing anymore. GPs were recruited by email and answered the survey without any incentive.

### Procedures

The questionnaire was published online in English and 21 other languages corresponding to the participating countries on SurveyMonkey® (www.surveymonkey.com, Palo Alto, CA, USA). Content validity of the translations where checked by the national coordinators who were all fluent in English. In Ukraine, where web access was limited, a paper version was administered. The first set of survey questions determined GP -specific characteristics (sex, years of experience as a GP, estimation of prevalence of oldest-old patients and location of their practice). Then, GPs were asked if they used hypertension guidelines to decide how to treat the oldest-old, and which guidelines they used. We defined the first guideline they mentioned as the most important. We analyzed all the documented guidelines and we categorized them. When local guidelines referred to another guideline (e.g., from the ESC), we counted it as the second guideline. If GPs listed something other than a guideline, we classified it under "Others" (S1 Appendix).

The complete survey described eight case-vignettes where oldest-old male or female patients presented for a routine control (Additional file 1 in [14]). These patients had no

symptoms suggesting hypertension and took no antihypertensive medication. The vignettes differentiated by the following variables: systolic blood pressure (SBP; options: 140 or 160mmHg), presence or absence of history of cardiovascular disease like myocardial infarction or stroke, and presence or absence of frailty. In each case, GPs were asked to decide whether they would start antihypertensive treatment. In our study, we analysed data from four of the eight case-vignettes that applied to frail oldest-old patients. We defined frailty when at least two of the following Fried's criteria were present: unintentional weight loss, muscle weakness, exhaustion, slow gait speed and low level of activity [17].

## Statistical analysis

We used descriptive statistics to compare baseline characteristics in the whole sample and stratified by guideline use (rather yes and yes = yes; neutral, rather no and no = no). We used a Chi2-test to assess categorical data and a complete case analysis to handle missing data. To assess how GPs varied in their use of guidelines when they treated hypertension in the oldest-old, we calculated the crude proportions and 95% confidence intervals (CI) per country. To assess the role that guidelines played in GPs' decisions, we used a mixed-effects Poisson model to calculate percentages and 95% CIs of GPs who decided to treat hypertension across the four case-vignettes. We adjusted the model for sex and years of experience and stratified it by guideline use. We used a mixed-effects model to account for a clustering effect within each country and used the same model to stratify the guidelines that GPs said they followed further. To lower the risk of selection bias in countries with a low response rate, we performed a sensitivity analysis restricted to countries where >60% of GPs responded. Based on the distribution of guidelines GPs mentioned (S1 Appendix), we made a 5-category group that included the three most frequently mentioned guidelines (ESC, NICE, NHG, and every other guideline), and GPs who said they did not use guidelines (reference group). We considered a two-sided p-value of 0.05 to be statistically significant. STATA 15.1 (StataCorp, College Station, TX, USA) was used for all analyses.

## Results

We received responses from the 29 following countries: Switzerland, France, Germany, Luxembourg, Netherlands, Italy, Spain, Portugal, United Kingdom, Ireland, Austria, Denmark, Norway, Sweden, Finland, Greece, Slovenia, Latvia, Czech Republic, Hungary, Macedonia, Turkey, Ukraine, Poland, Romania, Bosnia Herzegovina, Brazil, Israel and New Zealand. After excluding 42 GPs who were no longer practicing, we included 2,543 GPs. The median response rate across countries was 26% (Inter Quartile Range 10–62%). 52.7% of the participating GPs were women. About one third of GPs (37.6%) had more than 20 years of experience. The self-reported prevalence of oldest-old in most GP practices ranged from 10% to 20%. Only 7.2% listed a prevalence higher than 30%. Half of the GPs practiced in a city while the two remaining quarters of GPs were located in suburban and rural area.

About 60% of GPs mentioned using guidelines when they treat hypertension in the oldest-old. Female and GPs with under 20 years of experience were more likely to use guidelines while GPs with the most experience (more than 20 years) reported using them less frequently. We found that reported prevalence of seeing oldest-old patients and the location of GPs' practices were not significantly associated with guideline use (Table 1).

We found a large variation in guideline use across countries, ranging from less than 25% in New Zealand to almost 90% in Ukraine (Fig 1, S2 Appendix). Over 80% of GPs in Brazil, Greece, Czech Republic, Macedonia, Slovenia, Romania and Ukraine reported using guidelines. Across all countries, 20 different guidelines were mentioned. 95% of guideline users

**Table 1. Baseline characteristics of general practitioners by guideline use during decision-making on treatment of hypertension in oldest old patients (n = 2,543).**

| Characteristics | Overall 2,543 | Guideline use | | P-value[a] |
|---|---|---|---|---|
| | | Yes 1,510 (59.4%) | No 1,033 (40.6%) | |
| **Sex, n (%)** | | | | |
| Female | 1,341 (52.7) | 823 (54.5) | 518 (50.2) | 0.031 |
| **Clinical Experience, n (%)** | | | | |
| <5 years | 471 (18.5) | 314 (20.8) | 157 (15.2) | <0.001 |
| 5–10 years | 445 (17.5) | 274 (18.1) | 171 (16.5) | |
| 11–15 years | 341 (13.4) | 203 (13.5) | 138 (13.4) | |
| 16–20 years | 328 (12.9) | 204 (13.5) | 124 (12.0) | |
| >20 years | 956 (37.6) | 514 (34.1) | 442 (42.8) | |
| **Estimated prevalence of oldest-old, n (%)** | | | | |
| <10% | 851 (38.7) | 591 (39.2) | 260 (37.7) | 0.145 |
| 10–20% | 865 (39.4) | 576 (38.2) | 289 (41.9) | |
| 21–30% | 323 (14.7) | 222 (14.7) | 101 (14.7) | |
| >30% | 159 (7.2) | 120 (8.0) | 39 (5.7) | |
| **Location of the practice, n (%)** | | | | |
| City | 1292 (50.8) | 793 (52.5) | 499 (48.3) | 0.078 |
| Suburban | 599 (23.6) | 336 (22.3) | 263 (25.5) | |
| Rural | 651 (25.6) | 380 (25.2) | 271 (26.2) | |

[a] Chi-square test for categorical variables

mentioned at least one of the guidelines listed in S1 Appendix. Inherent to the distribution of number of participants per country, the most commonly mentioned guidelines were the

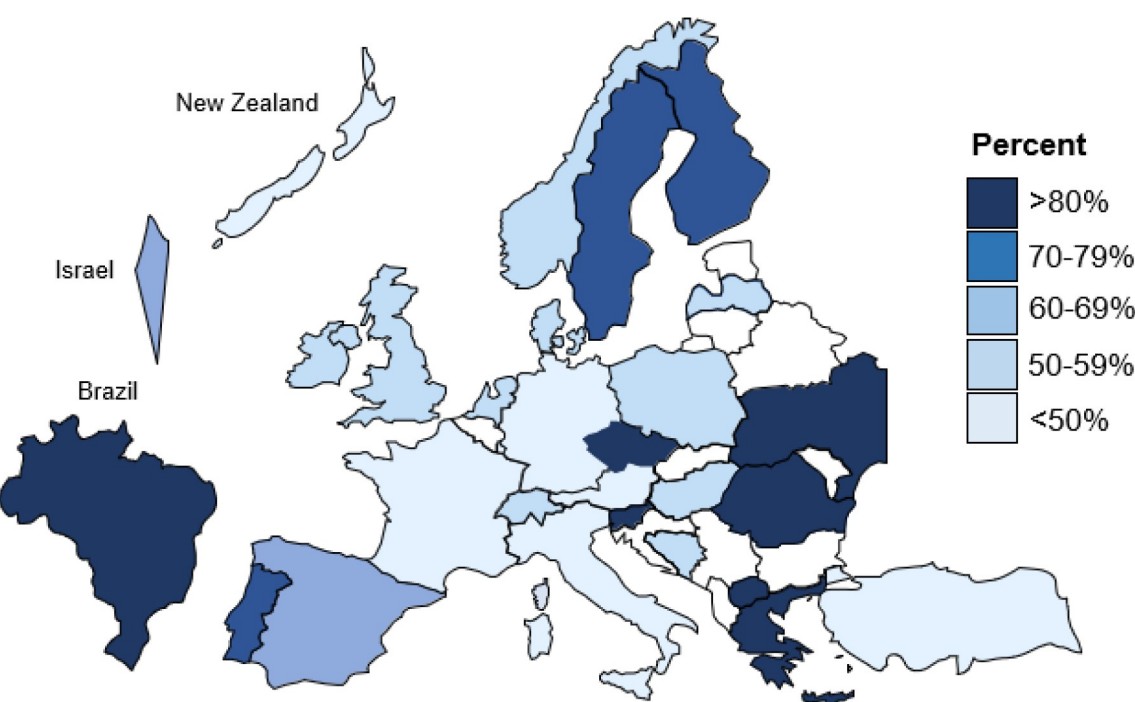

**Fig 1. Crude percentages of general practitioners using guidelines by country when treating hypertension in frail oldest-old.**

**Table 2. Proportions of general practitioners starting antihypertensive treatment in frail oldest-old stratified by history of cardiovascular disease, systolic blood pressure and use of guidelines.**

| | Patients with no history of cardiovascular disease | | Patients with history of cardiovascular disease | |
|---|---|---|---|---|
| | SBP 140 mmHg | SBP 160 mmHg | SBP 140 mmHg | SBP 160 mmHg |
| | GPs starting treatment (95% CI) | GPs starting treatment (95% CI) | GPs starting treatment (95% CI) | GPs starting treatment (95% CI) |
| **Guideline** | | | | |
| - Users | 16% (11%-24%) | 80% (69%-94%) | 37% (28%-49%) | 86% (77%-97%) |
| - Non-users | 12% (7%-18%) | 73% (61%-88%) | 31% (23%-43%) | 80% (69%-92%) |
| **P-value** | 0.015 | 0.13 | 0.09 | 0.12 |

Proportions and p-values comparing users and non-users from mixed Poisson regression models adjusted for GP sex, years of experience and country

NICE, ESC and NHG (the 'Dutch College of GPs') guidelines. They resulted in 60% of mentions. Treatment recommendations of these three guidelines are summarized in S3 Appendix. There were no specific treatment recommendations for frail oldest-old patients in these three guidelines regardless of the absence or presence of cardiovascular disease history. The decision was left to the treating physician.

Table 2 stratifies the treatment recommendation to start antihypertensive medication on guideline users and non-users. We found that proportions advising to start treatment differed by the case characteristics. However, GPs made similar decisions about treating or not treating hypertension in frail oldest-old patients, whether they reported to use guidelines or not. The exception in the case of a patient without history of CVD and SBP 140mmHg, in which there was an evidence for more treatment in guideline users (16% of guideline users decided to treat, 95%CI 11%-24%) compared to non-users (12% of non-users decided to treat, 95%CI 7–18%, p = 0.015). However, when restricting GPs to only those countries with a higher than 60% response rate (n = 8 countries; 676 participants), this difference was no longer statistically significant: guideline users and non-users (19%, 95%CI 9%-40%, p = 0.28).

In Fig 2, we further stratified GP treatment decisions by the three most often mentioned guidelines, other guidelines, and no guidelines. GPs in all categories made similar decisions for each case-vignette, no matter which guidelines was applied (or no guideline applied). However, there seems to be a trend that NHG-users were less likely to treat patients without history of CVD when SBP was 160 mmHg, but this finding was not statistically significant.

## Discussion

Our study of more than 2,500 GPs from 29 countries found that about 60% of GPs reported to use guidelines when treating hypertension in frail oldest-old. These proportions varied largely between countries, from less than 25% in New Zealand to almost 90% in Ukraine. Less experienced GPs and female GPs were more likely to use guidelines. However, GPs from all countries overall seemed to make similar treatment decisions when confronted with cases of frail oldest-old patients, whether or not they used guidelines, and regardless which guidelines they used.

### Clinical context and comparison with existing literature

While guideline use might have no major effect on treatment decisions in frail oldest-old, frailty, systolic blood pressure and history of cardiovascular disease seemed to have an influence [14]. In addition, country-specific factors such as cardiovascular burden and life expectancy are associated with the decisions when managing hypertension in this age group [15].

An explanation of why no association between guideline use and treatment decisions was found could be the absence of specific and clear recommendations in most current guidelines for this population group. As shown in S3 Appendix the decision to treat frail oldest-old patients with systolic blood pressure of 160 mmHg was left to the treating physician in the three most frequently mentioned guidelines at the time of the survey. Despite the lack of specific recommendation, the problematics of frail and older patients are discussed in these three guidelines. However no major changes were made concerning frail oldest-old in the new versions of ESC, NICE and NHG guidelines since 2016 and there are still no specific recommendations for this patient group [11–13]. Therefore, GPs are left to make treatment decisions based on other factors such as patient characteristics and personal judgement rather than on guidelines [18]. We assume GPs would use more guidelines if they were more applicable to the types of patients they treat. Moreover, the literature outlines patient safety to be more important than adherence to guidelines [19].

Some guidelines, however, pay a particular attention to oldest-old and frail patients e.g. the NHG guideline [13]. This work was an initiative of Dutch GPs involving all healthcare professionals in cardiovascular disease prevention in a multidisciplinary workgroup. In our study, in the case of primary prevention and SBP of 160mmHg, we could see that NHG-users seemed to treat less, however, the confidence interval overlapped with the proportions of GPs that adhered to other or no guidelines. This observation may imply that guidelines could influence GPs' treatments decision in frail oldest-old if specific recommendations are provided.

In the present study, we found that female doctors were more likely to use hypertension guidelines when treating frail oldest-old patients. This is in line with findings from other studies that described higher adherence to clinical guidelines by female physicians when treating other chronic conditions such as diabetes [20].

## Limitations and strengths

This study has several limitations but also strengths. First, we did not compare GPs' treatment decisions in oldest-old patients with and without frailty, which is why we cannot comment on

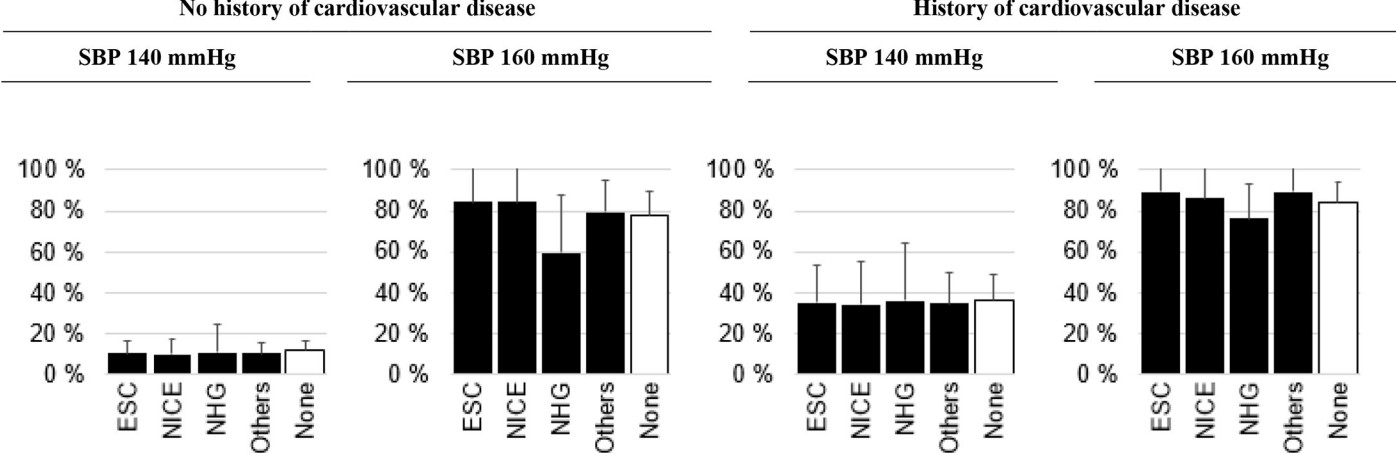

Proportions and p-values comparing users and non-users from mixed Poisson regression models adjusted for GP gender, years of experience and country

**Fig 2. General practitioners deciding to start antihypertensives in frail oldest-old stratified by type of guidelines used and no guideline used.**

the effect of guidelines on GPs' treatment decisions in non-frail oldest-old adults. Second, asking GPs what they would do is not the same as measuring what they really do. However, the use of a case-vignette study allows comparing decisions in different countries while still having a standardized situation, which can be seen as a strength when comparing across countries. We believe anonymity also lowered social desirability bias risk. Third, our median response rate of 26% is quite modest, but studies with GPs often have such a moderate response rate. In order to minimize and control for selection bias we mitigated that risk by running a sensitivity analysis of countries where the response rate was higher (more than 60%). We further acknowledge that due to different numbers of respondents per country the list of most used guidelines is skewed to overestimate responses from countries with many participants. However, we focused on the variety by including almost 30 countries, with some being able to recruit more and other being able to recruit less GPs. This approach also let us include responses from countries sometimes under-represented in research. Moreover, it is the first study to our knowledge to investigate guideline use and treatment decisions in frail and oldest-old with hypertension through standardized case-vignettes.

## Implications for research and/or practice

Until future primary care trials with oldest-old and frail patients will assess the benefits and risks of hypertension treatment in this population group, our study suggests that due to the remaining clinical dilemma, some GPs will choose not to follow any guidelines at all. One possible explanation may be the absence of specific recommendations for a highly heterogenous group: the oldest-old patients. Second, the absence of agreement between the various recommendations was found to be associated with a large variation in how GPs apply preventive measures [21]. The development of future guidelines should ideally address these differences and enhance the consensus with guidelines approved by all stakeholders. Further, guideline committees would benefit from sustained efforts in consulting patients as well as GPs in order to be able to better identify the needs of this patient group. Such an approach would decrease the potential of conflicting interests compared to guidelines written by professional societies and might lower the risk of overtreatment [22,23]. A good illustration of this kind of recommendations is the guideline from the NICE about the management of multimorbid patients published in 2016 [24]. The present guidelines with an often complex and ambiguous text might also be an important barrier to GPs' adoption of recommendations [25].

Future efforts should be made by including oldest-old patients into clinical studies and GPs in guideline committees to develop more specific guidelines with recommendations for oldest-old and frail patients with hypertension.

## Conclusions

Most GPs reported to be using guidelines when treating hypertension in oldest-old patients, but there was variation across countries and a plurality of different guidelines were mentioned. Nevertheless, guideline-users made similar treatment decisions compared to non-users. This suggests that the individual patient characteristics have a higher impact on GPs' treatment decisions than guidelines, which still fail to provide guidance concerning the optimal treatment in oldest-old and frail patients.

## Supporting information

**S1 Appendix. Names of the most frequent guidelines mentioned.**
(PDF)

**S2 Appendix. Percentage of guideline use per country.**
(PDF)

**S3 Appendix. Hypertension treatment recommendations for frail oldest-old and oldest-old in the three most mentioned guidelines.**
(DOCX)

**S1 Data.**
(DTA)

## Acknowledgments

The authors thank all the general practitioners who acted as national coordinators or participated in this study. They also thank Kali Tal for her editorial suggestions.

## Author Contributions

**Conceptualization:** Sven Streit.

**Data curation:** Céline Roulet.

**Formal analysis:** Céline Roulet, Zsofia Rozsnyai.

**Funding acquisition:** Céline Roulet.

**Investigation:** Céline Roulet.

**Methodology:** Céline Roulet, Zsofia Rozsnyai, Sven Streit.

**Project administration:** Céline Roulet.

**Resources:** Céline Roulet.

**Supervision:** Sven Streit.

**Validation:** Zsofia Rozsnyai, Katharina Tabea Jungo, Milly A. van der Ploeg, Carmen Floriani, Donata Kurpas, Shlomo Vinker, Sanda Kreitmayer Pestic, Kathryn Hoffmann, Rita P. A. Viegas, Christian Mallen, Athina Tatsioni, Hubert Maisonneuve, Claire Collins, Heidrun Lingner, Rosy Tsopra, Yolanda Mueller, Rosalinde K. E. Poortvliet, Sven Streit.

**Visualization:** Zsofia Rozsnyai, Ferdinando Petrazzuoli, Jacobijn Gussekloo, Sven Streit.

**Writing – original draft:** Céline Roulet.

**Writing – review & editing:** Zsofia Rozsnyai, Katharina Tabea Jungo, Milly A. van der Ploeg, Carmen Floriani, Donata Kurpas, Shlomo Vinker, Sanda Kreitmayer Pestic, Ferdinando Petrazzuoli, Kathryn Hoffmann, Rita P. A. Viegas, Christian Mallen, Athina Tatsioni, Hubert Maisonneuve, Claire Collins, Heidrun Lingner, Rosy Tsopra, Yolanda Mueller, Rosalinde K. E. Poortvliet, Jacobijn Gussekloo, Sven Streit.

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
