## [Decision Letter · Decision Letter 0]

5 May 2020

PONE-D-20-09925

Managing hypertension in frail oldest-old – the role of guideline use by general practitioners from 29 countries

PLOS ONE

Dear Prof. Dr. Streit,

Thank you for submitting your manuscript to PLOS ONE. After careful consideration, we feel that it has merit but does not fully meet PLOS ONE’s publication criteria as it currently stands. Therefore, we invite you to submit a revised version of the manuscript that addresses the points raised during the review process.

Two experts raised several points to revise.  Oldest-old without fraility control might be difficult to recruit and it is the limitation of this study.

We would appreciate receiving your revised manuscript by Jun 19 2020 11:59PM. To enhance the reproducibility of your results, we recommend that if applicable you deposit your laboratory protocols in protocols.io, where a protocol can be assigned its own identifier (DOI) such that it can be cited independently in the future. For instructions see: http://journals.plos.org/plosone/s/submission-guidelines#loc-laboratory-protocols

We look forward to receiving your revised manuscript.

Kind regards,

Tatsuo Shimosawa, M.D., Ph.D.

Academic Editor

PLOS ONE

Journal Requirements:

Reviewers' comments:

Reviewer's Responses to Questions

**Comments to the Author**

1. Is the manuscript technically sound, and do the data support the conclusions?

Reviewer #1: Partly

Reviewer #2: Yes

2. Has the statistical analysis been performed appropriately and rigorously? 

Reviewer #1: Yes

Reviewer #2: Yes

3. Have the authors made all data underlying the findings in their manuscript fully available?

Reviewer #1: Yes

Reviewer #2: Yes

4. Is the manuscript presented in an intelligible fashion and written in standard English?

Reviewer #1: Yes

Reviewer #2: Yes

5. Review Comments to the Author

Reviewer #1: In this study, the authors analyzed the data of the worldwide survey from GPs, and investigated whether guideline use influence the treatment choice for hypertensive oldest-old with frailty defined by Fried criteria. They found that GPs who used guidelines are less experienced and more female, but there was no specific tendency with respect to guideline use to treat frail oldest-old.

Major points

The authors speculate that guideline use had no effect on treatment decisions in frail oldest-old, because of the absence of specific and clear recommendations in most current guidelines for this population group. However, this speculation could suffer from the absence of positive control group in which guideline use significantly affects the treatment. Given that their questionnaires were limited to the treatment of the oldest-old, it might be helpful to assess the treatment of oldest-old with non-frailty and compare the results in those with frailty. The hypothesis can be strengthened, if there is any difference in treatment tendency between the guideline users and non-users in non-frailty patients.

The reviewer recommends to show the table to summarize the recommendation of treatment in oldest-old with or without frailty among clinical guidelines. If there is no specific recommendation to these populations in guidelines, the recommendation to treatment in older patients can be alternatively shown.

Reviewer #2: Dear Authors,

Thank you for submitting the research. The topic is important and the results are relevant.

Though the manuscript is written in a sufficient way, please find some minor suggestions for improvement:

- Abstract section Design/Section could be supplemented with some setting description;

- Abstract Methods section lists the countries that participated in the study: Europe, Brasil, Israel and New Zealand. But the first is continent and the others are countries. I suggest writing "European countries". By the way, I'd see it rational to mention the countries participating in the research.

- I suggest adding some explanation about the guidelines in the Introduction section. Are they national, international, similar in some countries, etc. Are they available in all countries?

- In the Setting section you mention EGPRN as an example. I suggest to include some other as well or not mention organization at all.

- Please describe how the GPs were selected to involve as the participants. Was the criterion(s) similar for all the countries or local coordinators decided? How their contacts for sending the questionnaire link were gathered?

- I suggest adding some statistics on GPs distribution among the countries. Maybe adding (if it's available) urban/rural distribution. I believe it would be valuable to present some more correlations between demographics and using the guidelines. Of course, if it's available.

6. PLOS authors have the option to publish the peer review history of their article (what does this mean?). If published, this will include your full peer review and any attached files.

Reviewer #1: No

Reviewer #2: No

---

## [Author Response · Author response to Decision Letter 0]

18 Jun 2020

1st Reviewer comments:

1. In this study, the authors analyzed the data of the worldwide survey from GPs and investigated whether guideline use influence the treatment choice for hypertensive oldest-old with frailty defined by Fried criteria. They found that GPs who used guidelines are less experienced and more female, but there was no specific tendency with respect to guideline use to treat frail oldest-old.

Major points

The authors speculate that guideline use had no effect on treatment decisions in frail oldest-old, because of the absence of specific and clear recommendations in most current guidelines for this population group. However, this speculation could suffer from the absence of positive control group in which guideline use significantly affects the treatment. Given that their questionnaires were limited to the treatment of the oldest-old, it might be helpful to assess the treatment of oldest-old with non-frailty and compare the results in those with frailty. The hypothesis can be strengthened, if there is any difference in treatment tendency between the guideline users and non-users in non-frailty patients.

Response: We thank the reviewer for pointing out this important aspect. We agree that the absence of positive control group with oldest old patients without frailty is a limitation of our study. We therefore clarified this point in the section “Limitations and strengths” on page 12. We consciously chose to focus on frail oldest-old because of two main reasons. First, recommendations for this specific population are missing from most hypertension guidelines. This fact is described in the literature [references 1 and 2 below] as well as in guidelines. For example, these two citations from the hypertension guideline of the European Society of cardiology (ESC) state that “all of the above recommendations relate to relatively fit and independent older patients, because physically and mentally frail and institutionalized patients have been excluded in most RCTs of patients with hypertension” and “In some patients, best achievable BP may be higher than the recommended target” [3]. Secondly, this population group is especially at risk for negative consequences of too aggressive blood pressure treatment while oldest-old without frailty may have a greater tolerance [4]. 

2.The reviewer recommends to show the table to summarize the recommendation of treatment in oldest-old with or without frailty among clinical guidelines. If there is no specific recommendation to these populations in guidelines, the recommendation to treatment in older patients can be alternatively shown.

Response: We appreciate this suggestion and we added a new table to summarize the main recommendations in the appendix S3. We chose to present the recommendations for frail oldest-old of the three most mentioned guidelines, which were effective at the time of our study in 2016. We hope this table will help to get an overview of the contents of the guidelines. We also provided the information on guidelines changes since then (refer to the Discussion Section on page 11). 

2nd Reviewers comments:

1. Dear Authors,

Thank you for submitting the research. The topic is important and the results are relevant.

Though the manuscript is written in a sufficient way, please find some minor suggestions for improvement:

- Abstract section Design/Section could be supplemented with some setting description;

Response: We completed this section with the following sentence: “Cross-sectional study among currently active GPs from 29 countries using a case-vignettes survey.”

2. Abstract Methods section lists the countries that participated in the study: Europe, Brazil, Israel and New Zealand. But the first is continent and the others are countries. I suggest writing "European countries". By the way, I'd see it rational to mention the countries participating in the research

Response: We thank the reviewer for this comment and adapted in the abstract as follows: “GPs from 26 European countries and from Brazil, Israel and New Zealand were invited”. We listed all the countries under “Results” on page 8. 

3. I suggest adding some explanation about the guidelines in the Introduction section. Are they national, international, similar in some countries, etc. Are they available in all countries?

Response: This comment is relevant and we agree that some more explanations about the different guidelines would help the reader. We chose to integrate a short description of the guidelines in the “Introduction” on page 5. 

4. In the Setting section you mention EGPRN as an example. I suggest to include some other as well or not mention organization at all.

Response: We agree with this remark and removed the name of this organization.

5. Please describe how the GPs were selected to involve as the participants. Was the criterion(s) similar for all the countries or local coordinators decided? How their contacts for sending the questionnaire link were gathered?

Response: We are happy to clarify this point. The GPs were contacted by a national coordinator, which we collaborate with through national and European organizations, as explained in the section “Methods” from page 5 to 7. To clarify the inclusion criteria, we adapted the subsection “Participants” on page 6 as follow: “The only inclusion criteria to participate in the ATTENTIVE study, was to be currently active as a GP”. National coordinators sent the link of the survey per e-Mail, except in Ukraine where a paper version was distributed as detailed on page 6 and 7. More explanations on the data collection can be found in the original publications of the ATTENTIVE study [5,6]. 

6. I suggest adding some statistics on GPs distribution among the countries. Maybe adding (if it's available) urban/rural distribution. I believe it would be valuable to present some more correlations between demographics and using the guidelines. Of course, if it's available.

We thank the reviewer for this relevant comment. Data about the location of GP’s practices were collected in the ATTENTIVE study. We therefore completed our sections “Methods” and “Results” as well as Table 1. We found no association between GPs’ geographical location and guideline use. 

References

1) Messerli FH, Sulicka J, Gryglewska B. Treatment of hypertension in the elderly. N Engl J Med. 2008;359(9): 972–3. author reply 973-974.

2) Roberts RG, Wynn-Jones J. Research and rural; EGPRN and EURIPA-finding common ground. October 2013, Malta. Eur J Gen Pract. 2015;21(1): 77-81

3) Williams B, Mancia G, Spiering W, et al. 2018 ESC/ESH Guidelines for the management of arterial hypertension. Eur Heart J. 2018;39(33):3021‐3104. 

4) Streit S, Poortvliet RKE, Gussekloo J. Lower blood pressure during antihypertensive treatment is associated with higher all-cause mortality and accelerated cognitive decline in the oldest-old – data from the Leiden 85-plus Study. Age and Ageing. 2018;47(4): 545-550.

5) Streit S, Verschoor M, Rodondi N, Bonfim D, Burman RA, Collins C, et al. Variation in GP decisions on antihypertensive treatment in oldest-old and frail individuals across 29 countries. BMC Geriatr. 2017;17(1): 93.

6) Streit S, Gussekloo J, Burman RA, Collins C, Kitanovska BG, Gintere S, et al. Burden of cardiovascular disease across 29 countries and GPs' decision to treat hypertension in oldest-old. Scand J Prim Health Care. 2018(1): 89-98.

---

## [Decision Letter · Decision Letter 1]

29 Jun 2020

Managing hypertension in frail oldest-old – the role of guideline use by general practitioners from 29 countries

PONE-D-20-09925R1

Dear Dr. Streit,

We’re pleased to inform you that your manuscript has been judged scientifically suitable for publication and will be formally accepted for publication once it meets all outstanding technical requirements.

Kind regards,

Tatsuo Shimosawa, M.D., Ph.D.

Academic Editor

PLOS ONE

Additional Editor Comments (optional):

Reviewers' comments:

Reviewer's Responses to Questions

**Comments to the Author**

1. If the authors have adequately addressed your comments raised in a previous round of review and you feel that this manuscript is now acceptable for publication, you may indicate that here to bypass the “Comments to the Author” section, enter your conflict of interest statement in the “Confidential to Editor” section, and submit your "Accept" recommendation.

Reviewer #1: All comments have been addressed

2. Is the manuscript technically sound, and do the data support the conclusions?

Reviewer #1: Yes

3. Has the statistical analysis been performed appropriately and rigorously? 

Reviewer #1: Yes

4. Have the authors made all data underlying the findings in their manuscript fully available?

Reviewer #1: Yes

5. Is the manuscript presented in an intelligible fashion and written in standard English?

Reviewer #1: Yes

6. Review Comments to the Author

Reviewer #1: (No Response)

7. PLOS authors have the option to publish the peer review history of their article (what does this mean?). If published, this will include your full peer review and any attached files.

Reviewer #1: No

---

## [Editor Report · Acceptance letter]

30 Jun 2020

PONE-D-20-09925R1 

Managing hypertension in frail oldest-old – the role of guideline use by general practitioners from 29 countries 

Dear Dr. Streit:

I'm pleased to inform you that your manuscript has been deemed suitable for publication in PLOS ONE. Congratulations! Your manuscript is now with our production department. 

Kind regards, 

on behalf of

Prof. Tatsuo Shimosawa 

Academic Editor

PLOS ONE